# Nuclear VANGL2 Inhibits Lactogenic Differentiation

**DOI:** 10.3390/cells13030222

**Published:** 2024-01-25

**Authors:** Stefany Rubio, Rut Molinuevo, Natalia Sanz-Gomez, Talieh Zomorrodinia, Chad S. Cockrum, Elina Luong, Lucia Rivas, Kora Cadle, Julien Menendez, Lindsay Hinck

**Affiliations:** 1Institute for the Biology of Stem Cells, University of California, Santa Cruz, CA 95064, USA; 2Department of Molecular, Cell and Developmental Biology, University of California, Santa Cruz, CA 95064, USA; 3Department of Cancer Biology, Institute for Biomedical Research “Alberto Sols”, 28029 Madrid, Spain

**Keywords:** VANGL2, mammary gland, nuclear localization

## Abstract

Planar cell polarity (PCP) proteins coordinate tissue morphogenesis by governing cell patterning and polarity. Asymmetrically localized on the plasma membrane of cells, transmembrane PCP proteins are trafficked by endocytosis, suggesting they may have intracellular functions that are dependent or independent of their extracellular role, but whether these functions extend to transcriptional control remains unknown. Here, we show the nuclear localization of transmembrane, PCP protein, VANGL2, in the HCC1569 breast cancer cell line, and in undifferentiated, but not differentiated, HC11 cells that serve as a model for mammary lactogenic differentiation. The loss of *Vangl2* function results in upregulation of pathways related to STAT5 signaling. We identify DNA binding sites and a nuclear localization signal in VANGL2, and use CUT&RUN to demonstrate recruitment of VANGL2 to specific DNA binding motifs, including one in the *Stat5a* promoter. Knockdown (KD) of *Vangl2* in HC11 cells and primary mammary organoids results in upregulation of *Stat5a*, *Ccnd1* and *Csn2*, larger acini and organoids, and precocious differentiation; phenotypes are rescued by overexpression of *Vangl2*, but not *Vangl2*_Δ*NLS*_. Together, these results advance a paradigm whereby PCP proteins coordinate tissue morphogenesis by keeping transcriptional programs governing differentiation in check.

## 1. Introduction

Planar cell polarity (PCP) refers to the alignment and organization of cells relative to the proximal–distal tissue plane [1]. A core suite of evolutionarily conserved proteins is responsible for translating global patterning information into asymmetric localization patterns that coordinate morphogenetic behaviors of cells within tissues. The asymmetric localization of these core PCP proteins depends on their differential trafficking that is highly regulated both from the trans-Golgi network following their synthesis, and from the plasma membrane following their internalization when they are redistributed during tissue morphogenesis and after cell division [2]. Recent studies have shown that, upon endocytosis, at least one PCP protein, Van Gogh-like protein 2 (VANGL2), is incorporated into the intracellular membranous compartment where it is involved in numerous activities. For example, a lysosomal function for VANGL2 has been identified in controlling chaperone-mediated autophagy that blocks osteogenesis by binding LAMP2A and preventing the selective degradation of OB-lineage-inhibiting factors [3]. There are also reports of VANGL2 in perinuclear endocytic vesicles in a variety of contexts including a breast cancer cell line, SKBR7 [4], hUVECs under low-fluid shear stress [5], in complexes with the cytoplasmic scaffold protein GIPC [6], and when induced by either the depletion of nucleocytoplasmic shuttling protein Dapper1 [7] or the Adaptor Protein-3 complex [8]. Taken together, these studies suggest that in addition to VANGL2’s key role at the plasma membrane as a PCP protein, it has yet undiscovered roles within the membranous compartments of the cell.

The mammary gland is an organ that undergoes serial stages of morphogenetic tissue remodeling. The initial stage occurs during embryonic development when a nascent ductal structure is formed. The second stage ensues during puberty when the mammary gland undergoes branching morphogenesis to form a mature ductal tree. The third stage takes place in adulthood when the mammary gland is primed for pregnancy with each menstrual cycle and then, with the onset of pregnancy, massively remodeled to build a milk supply in a process known as alveologenesis. We previously showed that VANGL2 is required for normal embryonic mammary gland development and branching morphogenesis. The loss of *Vangl2* function by either conditional knockout or by missense mutation (*Vangl2^Lp/Lp^*), which disrupts trafficking to the plasma membrane, results in several incompletely penetrant morphological phenotypes, including dilated ducts, consistent with disrupted PCP signaling. We also observed disorganized, exuberant, tertiary, alveolar-like structures in the nulliparous animal, suggesting a role for VANGL2 in regulating alveolar differentiation [9]. We identified significant downregulation of *B cell-specific Moloney murine leukemia virus integration site 1* (*Bmi1*) in both *MMTV-Cre*; *Vangl2^fl/fl^* mammary tissue and *Vangl2^Lp/Lp^* outgrowths. BMI1 is a core component of the Polycomb Repressive Complex 1 (PRC1) that, along with Polycomb Repressive Complex 2 (PRC2), functions in the nucleus as an epigenetic repressor through chromatin remodeling [10]. In the mammary gland, the loss of *Bmi1* results in the formation of premature alveolar-like structures in the nulliparous animal [11], echoing the phenotype observed in *Vangl2^Lp/Lp^* outgrowths [9], and raising the possibility that VANGL2 may also play a regulatory role in repressing transcription.

It is now well established that transmembrane proteins, including growth factor and G-protein coupled receptors, are trafficked into the nucleus where they participate in not only the regulation of specific target gene transcription, but also in a wide range of processes including DNA replication, DNA repair, and RNA metabolism [12,13]. Here, we identify DNA binding sites and a nuclear localization signal in VANGL2 and employ Cleavage Under Targets and Release Using Nuclease (CUT&RUN) to identify VANGL2 DNA binding motifs, one of which is found within the *Stat5a* promoter. We show that VANGL2 localizes to the nucleus of undifferentiated, but not differentiated, mammary epithelial cells. The loss of *Vangl2* results in upregulation of *Stat5a* and its target genes *Ccnd1*, *Csn2*, and *Wap*, along with precocious differentiation that is rescued by the overexpression of *Vangl2*, but not *Vangl2*_Δ*NLS*_. Together, these data identify a new role for VANGL2 in regulating lactogenic differentiation by acting as a transcriptional repressor of *Stat5a*.

## 2. Materials and Methods

### 2.1. Cells and Mouse Strain

Just prior to initiating the studies, we purchased HCC1569 cells (CRL-2330™) and HC11 cells (CRL-3062™) directly from ATCC (Manassas, VA, USA. HCC1569 cells were isolated from a metaplastic carcinoma patient. HC11 cells were derived from mammary glands of an inbred BALB/C strain mouse. We expanded the lines, froze numerous vials and then thawed new, low-passage cells to use for experiments and for generating the *Vangl2* KD lines. The HC11 cell line behaved as expected and differentiated over the published time course in response to dexamethasone, insulin, and prolactin. Outbred CD-1 mice were used for organoid culture experiments and obtained from Charles River Laboratories (Wilmington, MA, USA; 022CD1).

### 2.2. Plasmids

The lentivirus plasmids used to transduce cells have the different forms of *Vangl2* cloned into NheI/NotI digested pCDH-CMV-MCS-EF1-Puro. At the N-terminal of *Vangl2*, there is a fused 3XFLAG-GFP. The Vangl2_Δ*NLS*_ construct contains a span of mutations in the Nuclear Localization Signal (NLS) motif of VANGL2, amino acids 349-358 ERRVRKRRAR to EEEVEEEEAE. The plasmids were made by Epoch Life Science, Inc. (Sugarland, TX, USA).

### 2.3. CRISPR/Cas9

Two sgRNAs were designed for mouse *Vangl2* using Synthego (Redwood City, CA, USA) and validated using the UCSC genome browser. *Vangl2* was knocked out by Nucleofection (Lonza Biologics, Hayward, CA, USA: DS-138) HC11 cells with individual sgRNAs and 100 nM of homology directed repair (HDR) template containing stop codons in each frame and a restriction digest site (HindIII: AAGCTT) not found within *Vangl2* or *Cas9*. The HDR template was designed by spanning 50 ± bp from the PAM cut site, at which a stop codon 5′-TAACTAACTAA-3′ and a species-specific unique restriction digest site were added. The cells were grown in 24-well dishes until confluent before isolating into cells that were grown into single colonies in a 96-well dish, and mutants were validated by Western blotting. Two successful *KD* cell lines were used for experiments presented here. The relevant sequences are as follows: mV2sgRNA1 5′-CCCGGCUCUUAGAGCGGUGU-3′, mV2sgRNA2 5′-UAGAGCGGUGUCGGUCCCGG-3′, HDR Template seq1 5′-TTTGTTTCTGTTTCTCTGGCTTCCTGCTGCAGGGACCGCCGGGACGACATAACTAACTAAAAGCTTCCGCTCTAAGAGCCGGGATGGAG-TCGTGGAGATAAATCAGTGACGATCC-3′, HDR Template seq2: 5′-CTCTTGACCTTTGTTTCTGTTTCTCTGGCTTCCTGCTGCAGGGACCGCCGTAACTAACTAAAAGCTTGGACCGACACCGCTCTAAGAGCCGGGATGGGAGTCGTGGAGATAAATCAG-3′.

### 2.4. Cell Culture and Immunocytochemistry

Cells: HCC1569 cells were grown in RPMI-1640 (Thermo-Fisher Scientific, Waltham, MA, USA; 72400047) supplemented with 10% FBS (Thermo-Fisher Scientific, Waltham, MA, USA; MT35010CV) and 1× Anti-Anti (Thermo-Fisher, Waltham, MA, USA; 15240112). Undifferentiated HC11 cells were cultured in RPMI-1640, supplemented with 10% FBS, 5 µg/mL insulin (Millipore-Sigma, Burlington VT, USA; I6634), 10 ng/mL epidermal growth factor (EGF; Thermo-Fisher Scientific, Waltham, MA, USA; AF-100-15), and 1× AntiAnti at 37 °C with 5% CO_2_. To differentiate HC11 cells, first the cells were made competent by growing them to confluency and then maintaining them in growing medium for 2 days. Next, the cells were primed by culturing them in priming medium [RPMI-1640 supplemented with 5% charcoal-stripped-FBS (Equitech Bio, Kerrville, TX, USA; SFBM31) 5 µg/mL insulin, 1 µM dexamethasone (Millipore-Sigma, Burlington VT, USA; D4902-1G), and 1× Anti-Anti] for 18 h at 37 °C with 5% CO_2_. To induce differentiation, primed HC11 cells were cultured in DIP Medium [RPMI-1640, supplemented with 10% FBS, 1 µM dexamethasone, 5 µg/mL insulin, 3 µg/mL Prolactin (National Hormone and Pituitary Program, Los Angeles, CA, USA; oPRL-21) and 1× Anti-Anti] at 37 °C with 5% CO_2_. HC11 and HCC1569 cells were seeded onto an 8-well chamber slide, which was coated with Poly-L-Lysine (Millipore-Sigma, Burlington VT, USA; P8920) for 30 min at room temperature followed by 2X DPBS washes, grown overnight at 37 °C, 5% CO_2_. Cells were fixed using 4% paraformaldehyde for 10 min on ice, permeabilized for 10 min on ice using PBS (Phosphate Buffered Saline) + 0.25% Triton-X 100 (Millipore-Sigma, Burlington VT, USA; X100), incubated in blocking buffer (PBS/1% normal donkey serum (NDS, Millipore-Sigma, Burlington VT, USA; D9663)/O.3M glycine) for 1 h, and incubated overnight at 4 °C with primary antibodies in blocking buffer. Slides were washed 3X with PBS/1% NDS and incubated with secondary antibodies (Thermo-Fisher, Waltham, MA, USA) for 1 h at room temperature, washed again 3X with PBS/1% NDS, and lastly with PBS containing Hoechst33342 (AnaSpec, Fremont, CA, USA; AS-83218) at a 1:2000 dilution. Cells were mounted with Fluoromount G (Southern Biotech, Homewood, AL, USA; 0100-01). Acini: HC11 cells were grown in 3D as previously described [14]. Briefly, 5000 HC11 cells were embedded in 10% Matrigel (Corning, Somerville, MA; CB-40230C) atop a denser base layer of 50% Matrigel (on-top method). Cells were grown in 3D for 10 days before fixation. The cell media were changed on the cells every 2 days, and the cells were fixed for immunofluorescent analysis by fixing the whole cultures in each well of an 8-well chamber slide and stained as described above. Primary Mosaic Organoids: Organoids from primary cells harvested from CD1 mice were generated as previously described [14]. Briefly, the differentially trypsinized cells were independently transduced with lentivirus constructs at 30 MOI for 72 h prior to their embedding in Matrigel as described above for 3D HC11 acini. Embedded primary cells were cultured in basal DMEM/F12 media (Thermo-Fisher, Waltham, MA, USA; 11039021) supplemented with 1X N-2 (Thermo-Fisher, Waltham, MA, USA; 17502048), 1X B-27 (Thermo-Fisher, Waltham, MA, USA; 12587010), 100 ng/mL Nrg1 (R&D Systems Minneapolis, MN, USA; 5898-NR-050), 42.5 ng/mL R-spondin (Thermo-Fisher, Waltham, MA, USA; 120-38), 1 nM Rho-inhibitor (Y-27632; Cell Signaling Technologies, Danvers, MA, USA; 13624S), and 10 ng/mL EGF. After 5 days, growing organoids were cultured in differentiation media (DMEM/F12 containing 1X N-2, 1X B-27, 100 ng/mL Nrg1, 42.5 ng/mL R-spondin, 1 nM Rho-inhibitor, 1 µg/mL prolactin, 5 µg/mL dexamethasone, and 5 µg/mL insulin) for an additional 3 days. Nuclei: Nuclei were isolated from HC11 using the EZ Nuclei Kit and mounted onto positively charged slides before fixing using 100% MeOH (totaling 50% MeOH fix) for 2 min at room temperature. A second fixation step was performed using 100% MeOH for 2 min. The nuclei were incubated with blocking buffer (1% Normal Donkey Serum in PBS + 0.3 M glycine) for 1 h. Primary antibodies were incubated in blocking buffer overnight at 4 °C in a humid chamber. The nuclei were washed 2X using PBS + 0.1% Triton X-100 for 5 min. Secondary antibodies (Thermo-Fisher, Waltham, MA, USA) were added for 1 h at room temperature in a humid chamber. The slides were washed 2X using PBS + 0.1% Triton X-100 for 5 min, and a third wash using Hoechst 33342 in PBS for 5 min. The nuclei were mounted in Fluoromount G and dried overnight before sealing with clear nail polish. Cells and organoids were using Zeiss Axio Imager Microscope. Nuclei were imaged on a Solamere Spinning Disk (Yokogawa, Houston, TX, USA; CSU-X1 spinning disk) confocal.

Primary antibodies: anti-VANGL2 (Millipore-Sigma, Burlington VT, USA; ABN2242), anti-BMI1 (Cell Signaling Technologies, Danvers, MA, USA; D42B3), anti-LMNB1 (Thermo-Fisher, Waltham, MA, USA; 33-2000), anti-STAT5 (Santa Cruz Biotech, Dallas, TX, USA; sc-836), anti-ELF5 (Thermo-Fisher, Waltham, MA, USA; 720380), anti-CCND1 (Cell Signaling Technologies, Danvers, MA, USA; 2978), anti-CSN2 (ABclonal, Woburn, MA, USA; A12749), anti-CSN2 (ABclonal, Woburn, MA, USA; A12749), anti-GFP (Aves Lab, Davis, CA, USA; GFP-1020), anti-KRT14 (Covance, San Carlos, CA, USA; PRB-155P), anti-KRT8 (Developmental Studies Hybridoma Lab, Iowa City, IA, USA; TROMA-1). Actin was visualized using phalloidin-conjugated with Alexafluor 488 (Abcam, Waltham, MA, USA; 176753).

### 2.5. Western Blotting

Subcellular fractionation was done as reported previously [15], with minor modifications: Hexaylene Glycol (Millipore-Sigma, Burlington, VT, USA; 112100) was substituted for 10% Glycerol in the buffers and 3 washes were performed in between each fraction using respective buffers. Protein expression was determined by running the lysates dissolved in equal volume of 2X Laemli Buffer on an SDS-Page gel for 1.5 h at 100 V. The proteins were transferred to a Immobilon-P Membrane (Millipore-Sigma, Burlington, VT, USA; IPVH85R) membrane for 1.5 H at 400 mA. The membranes were rinsed in TBST-Low Salt (0.15 M NaCl) before they were blocked in 5% BSA (VWR, Radnor, PA, USA; 103219-864) diluted in TBST-Low Salt and rocked at room temperature for 1 h. Primary antibody (in 5% BSA/TBST-Low Salt) incubation was done overnight on a rocking platform at 4 °C. Membranes were washed 3X using TBST-High Salt (0.4 M NaCl) and 1x in TBST-Low Salt, each for 5 min, rocking at room temperature. Membranes were then incubated with secondary antibodies conjugated with Horseradish Peroxidase (Jackson ImmunoResearch Labs, West Grove, PA, USA), diluted in 5% non-fat dry milk/TBST-Low Salt rocking at room temperature for 45 min. The membranes were washed 3X using TBST-High Salt and 1X TBST-Low Salt, each for 5 min, rocking at room temperature. Immunoblots were developed using Clarity ECL (BioRad, Hercules, CA, USA; 1705060) and detected using the BioRad ChemiDoc MP Imaging System (BioRad; 12003153). Protein expression was determined relative to a housekeeping protein specific to each cellular compartment probed. Primary antibodies: anti-pVANGL2-S79/S82/S84 (Abclonal, Woburn, MA, USA; AP1207), anti-VANGL2 (Millipore-Sigma, Burlington VT, USA; MABN750), anti-TUBB1 (Millipore-Sigma, Burlington VT, USA; T7816), anti-HH3 (Santa Cruz Biotech, Dallas, TX, USA; sc517576), anti-GAPDH (Santa Cruz Biotech, Dallas, TX, USA; sc-365062).

### 2.6. RNA-Sequencing

HC11 cells were grown in a 60 mm dish until 90% confluent and harvested using Trizol Reagent (Thermo-Fisher, Waltham, MA, USA; 15596026). The RNA extraction was done as described [16]; however, samples were not flash frozen because mammalian cells easily lyse. RNA libraries were generated using the NEBNext^®^ Ultra^TM^ RNA Library Prep Kit for Illumina^®^ (New England Biolabs, Ipswich, MA, USA; E7530). In brief, 1 µg of total RNA from *WT* and *Vangl2 KD1* and *KD2* HC11 cells was poly-A-selected and used to generate cDNA libraries, which were paired-end sequenced on a NovaSeq6000 platform for 50 cycles (50PE). Raw sequence reads were quality filtered (reads with an average <Q20 were discarded) and adapters were trimmed using fastp [17]. The processed reads were mapped to the GRCm38 mouse genome using HISAT2 [18]. Reads were counted over exons annotated in gencode’s GRCm38 v23 GTF annotation file at the gene level using featureCounts from subread [19]. Differential expression analysis was performed using those read counts and DEseq2 (Wald tests) [20]. Genes with an FDR (Benjamini–Hochberg-corrected *p*-value) and an absolute fold change > 2 were qualified as differentially expressed. Pathway enrichment was performed on sets of differentially expressed genes using ENRICHR [21].

### 2.7. CUT&RUN Data Processing and Analysis

We collected 100,000 cells from three independent dishes of HC11 cells at undifferentiated, confluent, and differentiated states. CUT&RUN was performed on HC11 cells using a rabbit anti-VANGL2 antibody (1:50, Millipore-Sigma, Burlington, VT, USA; ABN2242) and isotype control IgG (Cell Signaling Technologies, Danvers, MA, USA; 2729) and the CUT&RUN Assay Kit (Cell Signaling Technologies, Danvers, MA, USA; 86652). The resulting DNA fragment samples were generated into libraries using the SimpleChIP^®^ ChIP-seq DNA Library Prep Kit for Illumina^®^ (Cell Signaling Technologies, Danvers, MA, USA; 56795) and SimpleChIP^®^ ChIP-seq Multiplex Oligos for Illumina^®^ (Dual Index Primers) (Cell Signaling Technologies, Danvers, MA, USA; 47538). Samples were sequenced using the NovaSeq 6000, 50PE (UC Berkeley, Berkeley, CA, USA). Sequences were subjected to quality control using FastQC [22], then aligned to the mm39 mouse genome assembly with bowtie2 [23], followed by spike-in normalization. Duplication was assessed and removed when necessary, using Picard MarkDuplicates “http://broadinstitute.github.io/picard/” (accessed on 9 December 2023), and best quality reads according to bowtie2 quality score were filtered. Peaks were called using MACS [24], and annotated to genomic features using Chipseeker [25]. Replicate reproducibility was assessed with the Euclidean distance and plotted in a heatmap. DeepTools2 was used to plot the distribution of peaks relative to TSSs and profile the signal intensity [26]. Motif finding was performed with MEME and STREME [27], using the bed files and the JASPAR motif database as input. Integrative Genomics Viewer (IGV) was used for visualization of the peaks [28]. Enrichr was used for pathway enrichment analysis [29]. To identify the VANGL2 binding motifs in *Stat5a*, we obtained overlapping peaks from the undifferentiated samples using bedtools [30]. The discovery of representative motifs was performed using MEME with the following specifications: a maximum of 25 motifs from 5 to 30 nucleotides. The motifs obtained were compared with the peak sequences found in the *Stat5a* promoter. Among the representative motifs, we found one present in the 3 replicates of the *Stat5a* promoter.

### 2.8. Statistics

The mean ± SEM is reported. *p*-values are determined using unpaired *t*-test with a Welch’s correction, unless specified. Significance is denoted as *p* ≥ 0.05 = ns, 0.05 < *p* < 0.01 = *, 0.01 < *p* < 0.001 = **, and *p* < 0.001 = ***.

## 3. Results

### 3.1. VANGL2 Localizes to the Nucleus in HCC1569 Cells and Undifferentiated, but Not Differentiated, HC11 Cells

To investigate the subcellular localization of VANGL2, we initially examined HCC1569 cells, a highly proliferative, HER2+ breast cancer cell line that expresses high levels of *Vangl2* (Appendix A) [31,32]. We observed VANGL2 puncta in the nucleus that is outlined by nuclear envelope marker Lamin B1 (LMNB1) staining (Appendix A, arrow). We also observed punctate VANGL2 staining along the plasma membranes outlined by CDH1 staining (Appendix A, arrowhead). Because VANGL2 is not known to be localized in the nucleus, we further examined its subcellular localization by fractionation followed by immunoblotting (Appendix A). Using an antibody directed against three serines (S79/S82/S84) of VANGL2 that are phosphorylated by the ROR2 tyrosine kinase receptor (pVANGL2) and are reported to protect it from endoplasmic reticulum associated degradation [33,34,35], we found pVANGL2 present in both the cytoplasmic/membrane and nuclear fractions (Appendix A). We also used an antibody targeting the N-terminus of VANGL2 and detected at least four VANGL2 forms of varying phosphorylation with more hyperphosphorylated VANGL2 in the cytoplasmic/membrane fraction and less hyperphosphorylated VANGL2 in the nuclear fraction (Appendix A).

To explore a potential functional role for VANGL2 in the nucleus, we turned to HC11 cells because they offer an in vitro model of mammary lactogenic differentiation. To evaluate levels of *Vangl2* over the time course of differentiation, we interrogated a recently published RNA-seq dataset and found that *Vangl2* mRNA is present over the time course of differentiation, but its expression in undifferentiated HC11 cells is less than its expression in embryonic stem cells, and *Vangl2* levels decrease with differentiation (Appendix A) [36]. Next, we used immunocytochemistry to examine the subcellular localization of VANGL2 over the time course of HC11 cell differentiation. HC11 cells were grown to confluence and maintained for two days before priming by withdrawing epidermal growth factor. Differentiation was then induced by treating the cultures with prolactin, along with dexamethasone and insulin, resulting in the formation of milk domes in five to seven days (Figure 1A–D) [37]. Phase contrast images showed examples of cells at confluence and priming, and at differentiation day 7 when milk domes were visible (Figure 1B–D, top row). Immunostaining for VANGL2 (Figure 1B–D, middle row) and BMI1 (Figure 1B–D, bottom row) in confluent cells revealed strong nuclear staining that was slightly diminished at priming (Figure 1C). However, by seven days of differentiation, there was little to no nuclear VANGL2 and it appeared, instead, primarily as puncta in the cytoplasm (Figure 1D, middle row). BMI1 staining was largely absent at this stage (Figure 1D, bottom row). These data suggest that VANGL2 and BMI1 are in the nucleus in undifferentiated confluent HC11 cells, but that differentiation results in the loss of VANGL2 nuclear localization and diminished BMI1 expression. To further examine the nuclear localization of VANGL2, we isolated nuclei from confluent HC11 cells and immunostained for VANGL2 and BMI1, which were present in a largely non-overlapping pattern within the nucleus (Figure 1E). Examining VANGL2 with LMNB1, we found VANGL2 puncta overlapping with LMNB1 staining (Figure 1E, bottom row arrows).

We further investigated the subcellular localization of VANGL2 by fractionating confluent (C), primed, (P) and differentiated (D) HC11 cells, and immunoblotting for VANGL2 (Figure 1F–I). Using the anti-phosphoVANGL2 antibody, we observed pVANGL2 at every time point in all fractions: cytoplasmic/membrane, nuclear, and insoluble nuclear (Figure 1F,G). However, consistent with the immunocytochemical results, the nuclear fraction contained less pVANGL2 at priming and little to no pVANGL2 at differentiation. Concordantly, pVANGL2 increased in the cytoplasmic/membrane fraction at differentiation; however, in the insoluble nuclear fraction no changes were observed in the amount of pVANGL2 (Figure 1F,G). We also used the anti-N-terminus VANGL2 antibody and detected at least four VANGL2 forms of varying phosphorylation with different forms present in the three fractions (Figure 1H,I). At all stages of the time course, VANGL2 was present in the cytosolic/membrane fraction. VANGL2 was also in the nuclear and insoluble nuclear fractions in confluent and primed cells. However, in differentiated cells, there was little to no VANGL2 in the nuclear fraction and trending less in the insoluble nuclear fraction, while it increased in the cytoplasmic/membrane fraction (Figure 1H,I). Taken together, these data support the notion that nuclear localization of VANGL2 is dependent on the differentiation state of HC11 cells.

### 3.2. Undifferentiated Vangl2 Knockdown Cells Display Increased Polyploidy and Express Genes Involved in Lactogenic Differentiation

To determine the role of VANGL2 in HC11 cells, we targeted *Vangl2* using CRISPR/Cas9 and two different RNA guide sets. Two independently targeted cell lines were generated, and immunoblotting showed little to no VANGL2 in line 1 (*KD1*) and greatly diminished VANGL2 in line 2 (*KD2*) (Figure 2A and Appendix A). We examined the phenotype of these KD lines by growing wildtype (*WT*) and *KD* cells to confluence and staining for actin and DNA. We found that in contrast to the uniformly small nuclei of *WT* cells, the *Vangl2 KD* lines had cells with large nuclei that appeared to be polyploid (Figure 2B, arrows). Polyploidy is one of the characteristics of differentiated, milk-producing alveolar cells [38,39]. To further investigate whether HC11 cells undergo polyploidization with the loss of *Vangl2*, we performed FACS analysis (Figure 2C–E; Appendix A). In contrast to *WT* cells, cells from both *KD* lines had high scatter parameters, reflecting an increase in cell size and complexity (Figure 2C,D). Furthermore, DNA content analysis revealed an increased proportion of polyploid (>4C) cells in the *KD* lines relative to the *WT* (2.72-fold and 2.66-fold increase in *KD1* and *KD2*, respectively) (Figure 2E and Appendix A). In addition, we noted an increase in the proportion of S phase cells in *KD1*, suggesting this line was more proliferative than *KD2* (Appendix A).

To further examine the consequences of *Vangl2* loss, we performed RNA-seq analysis on confluent *WT* cells and the two different *Vangl2 KD* HC11 cell lines (Appendix A). Comparing the two *KD* lines to parent HC11 cells, we observed differential gene expression between each cell line, characterized by upregulated and downregulated genes (Appendix A). These differences in gene expression between the two *KD* lines were not surprising given that HC11 cells comprise a heterogenous population and, accordingly, the generation of *KD* lines from cell cloning resulted in lines with different transcriptomes. However, we reasoned that gene expression changes shared between the two lines, in relation to *WT* gene expression, could provide insight into the consequences of *Vangl2* loss (Appendix A). We performed pathway enrichment analysis on these overlapping genes using Enrichr and discovered shared upregulation of STAT5, IRF8, and IRF1 pathways (Appendix A) [21]. STAT5 signaling, a master regulator of lactogenic differentiation, is activated downstream of prolactin signaling and culminates in expression of milk protein genes, including beta-casein (*Csn2*) and whey acidic protein (*Wap*), both of which contain STAT5-responsive elements in their promoters [40]. STAT5 signaling also promotes proliferation through its upregulation of *Cyclin D1* (*Ccdn1*) [41]. IRF1 and IRF8 are downstream targets of JAK/STAT [42,43], and associated with milk production [44]. We further saw a very strong, shared downregulation of polycomb repressive complex (PRC) 1 and PRC2 pathway components SUZ12, EZH2, MTF2, and RNF2 (Appendix A). BMI1, which we previously showed is downregulated with the loss of *Vangl2* [9], is known to work with other PRC1 proteins to regulate mammary cell gene expression, promote mammary stem cell renewal, and prevent premature alveologenesis [11].

The discovery that *Vangl2 KD* lines had an increased proportion of polyploid cells and express higher transcript levels of genes involved in STAT5 signaling suggested these cells were undergoing precocious differentiation in the absence of differentiation media. To examine whether the pathways controlling lactogenic differentiation were being activated in *Vangl2* KD cells, we performed immunocytochemistry on confluent *WT* cells and the two *Vangl2 KD* cell lines (Figure 2F–K). Using a polyclonal antibody that recognizes both STAT5A and STAT5B, we found increased expression of these transcription factors (STAT5) in a subset of cells (Figure 2F,G) [45,46]. In addition, we observed increased expression of STAT5A target, ELF5, which was upregulated by STAT5 during pregnancy (Figure 2H,I) [46,47]. Furthermore, the expression of STAT5A target, CCND1, also robustly increased, suggesting that the loss of *Vangl2* increased proliferation (Figure 2J,K) [41]. To evaluate measures of lactogenic differentiation, we treated cells with DIP for 2 days and stained for F-actin to quantify milk domes (Appendix A) and for the milk protein CSN2 (Appendix A). In this short differentiation timeframe, both *Vangl2 KD* HC11 lines generated more milk domes and contained more CSN2 compared to *WT* cells (Appendix A). Thus, the loss of *Vangl2* resulted in increased STAT5A signaling and precocious differentiation.

### 3.3. VANGL2 Is Recruited to DNA Motifs in Undifferentiated, but Not Differentiated, HC11 Cells, Including One in the Stat5a Promoter

Our studies using *Vangl2 KD* cell lines did not directly address the nuclear function of VANGL2. Proteins that translocate to the nucleus frequently contain DNA binding sites and at least one nuclear localization signal (NLS). To search for DNA binding sites in VANGL2, we analyzed the mouse and human VANGL2 protein sequences using computational protein prediction tools (e.g., by DNAbinder, DRNApred, DP-Bind, and DisoRDPbind) and identified seven putative DNA binding sites, consisting of a series of positively charged amino acids in the N-terminal region of Vangl2 and two in the C-terminal region (Figure 3A). A putative monopartite NLS has also been identified in zebrafish VANGL2, which we confirmed in both mice and humans using cNLS mapper, locating it in the C-terminal end between amino acids 349–358 [48,49]. Additionally, closer to the C-terminus are two PDZ domains, an internal unconventional PDZ domain where the *Looptail* mutation (S464N) maps, and a typical class I domain [50]. Both these domains are interacting sites for other PCP proteins and PDZ domain-containing proteins that aid in VANGL2’s transport and signaling [51].

One way that nuclear VANGL2 could regulate gene expression is to directly bind DNA. To investigate, we performed CUT&RUN for VANGL2 at two stages of HC11 cell differentiation: confluent, when VANGL2 is in the nucleus (Figure 1B,C,E–I), and differentiated, when we observe little to no nuclear VANGL2 (Figure 1D,F–I). Pearson correlation showde the degree of similarity in VANGL2 DNA association between undifferentiated and differentiated HC11 cells, as well as an undifferentiated non-specific IgG control (Appendix A). Importantly, the differentiated HC11 cells had a higher degree of similarity with the non-specific IgG control than with undifferentiated HC11 cells (Appendix A), which is consistent with little to no VANGL2 in the nucleus during differentiation and in agreement with our previous observations (Figure 1D,F–I). We identified 31,062 VANGL2 binding sites in undifferentiated HC11 cells by detecting highly confident, overlapping peaks. Notably, ~29% (n = 8980) of the VANGL2 peaks were located within promoter regions (±1 kb from transcriptional start sites), ~35% within distal intergenic regions, and ~30% within intronic regions (Figure 3B,C). After annotating the peaks, we found they were associated with several pathways (Figure 3D), most notably E2F1 and RUNX2, which are both involved in the regulation of mammary stem/progenitor cells that need to be recruited and expanded to generate the prodigious cell growth occurring during pregnancy [52,53]. Other pathways, such as those governing basal cells and Rho GTPases, may reflect VANGL2’s role in PCP [51]. By performing motif analysis [27], we discovered the majority of peaks coincided with previously described DNA binding motifs for transcriptional regulators CTCF, RUNX2, and ELF5 (Figure 3E). CTCF organizes chromatin by binding DNA in intergenic regions where we also identify VANGL2 binding [54]. Furthermore, at the *Stat5a* locus, which contains two promoters [55], we find a strong peak at the second promoter, suggesting VANGL2 directly regulates *Stat5a* expression (Figure 3F and Appendix A). A repressive role for VANGL2 at this location in undifferentiated HC11 cells is consistent with our observation that *Vangl2* loss results in the upregulation of *Stat5a* and its target genes (Figure 2F–K and Appendix A).

### 3.4. Nuclear VANGL2 Inhibits HC11 Cell Proliferation, Acini Formation, and Expression of Stat5a and Its Target Genes Csn2 and Wap

To test the nuclear function of VANGL2 in repressing *Stat5a* expression, we mutated its NLS (Figure 4A), generating a construct (*V2*_Δ*NLS*_) that we overexpressed in undifferentiated *Vangl2 KD* HC11 cells using lentivirus. As controls, we overexpressed a *WT Vangl2* construct (*V2*) or a scrambled construct (*Scr*) (Figure 4B). To evaluate the phenotype in three-dimensional (3D) space, we plated these *KD* cells, along with *WT* HC11 cells, in Matrigel and allowed them to grow as mammary acini for 5 days before fixing and immunostaining them for the basal marker Keratin-14 (KRT14), with nuclei labeled by Hoechst (Figure 4B); we assessed lentiviral transduction by labeling for GFP (Appendix A). Consistent with elevated levels of CCND1 in the *KD* lines (Figure 2J,K), we observed a significant increase in the acini size of *Vangl2 KD1* compared to the *WT Scr* control, with a trending increase in *Vangl2 KD2* (Figure 4B,C), which expresses a higher level of VANGL2 (Appendix A). In both *Vangl2 KD* lines, *Vangl2* rescued this increase in size, generating acini that were *WT* in size or smaller (Figure 4B,C). *Vangl2*_Δ*NLS*_ overexpression, however, did not reduce acini size, suggesting the increased proliferation and acini size observed in *Vangl2 KD* cells was due to nuclear VANGL2 function (Figure 4B,C). The finding that *Vangl2*, but not *Vangl2*_Δ*NLS*_, overexpression rescuesd the *KD* phenotype was also observed when examining the expression of *Stat5a* at growing day 5 (Figure 4D). Consistent with VANGL2 acting as a transcriptional repressor in undifferentiated HC11 cells, *Stat5a* was upregulated in both *KD* cell lines, with *Vangl2* reducing *Stat5a* to below *WT* levels while *Vangl2*_Δ*NLS*_ had no effect (Figure 4D).

To further evaluate the nuclear function of VANGL2 in regulating the precocious differentiation of *Vangl2 KD* lines, we differentiated acini that were overexpressing either *Scr*, *Vangl2*, or *Vangl2*_Δ*NLS*_ for 2 days before harvesting and analyzing the expression of STAT5A target genes *Csn2* and *Wap* (Figure 4E,F). We found that both *Csn2* and *Wap* were upregulated in the *Vangl2 KD* clones (Figure 4E,F), consistent with the increase in milk domes and CSN2 observed in two-dimensional culture (Appendix A). Again, *Vangl2* reduced the expression of these genes to *WT* levels, whereas overexpression of *Vangl2*_Δ*NLS*_ did not. Taken together, our studies suggest that nuclear VANGL2 keeps the differentiation of HC11 cells in check by repressing the expression of master transcriptional factor STAT5A.

### 3.5. Nuclear VANGL2 Inhibits Mammary Luminal Cell Proliferation, Organoid Formation, and Expression of Stat5a and Its Target Genes Ccnd1 and Csn2

To further validate our discovery of a nuclear function for VANGL2, we grew primary murine mammary epithelial cells (MECs) as organoids in 3D Matrigel culture. We previously showed that the loss of *Vangl2* in the basal versus luminal compartments of the mammary gland resulted in different phenotypes [9]. The basal loss of *Vangl2* led to small organoids, whereas luminal loss resulted in larger organoids, similar to our observation in *Vangl2 KD* HC11 cells (Figure 4B,C). To examine whether the loss of nuclear *Vangl2* in the luminal compartment of primary organoids results in larger structures, we knocked down *Vangl2* in primary mammary luminal cells and then overexpressed either *Scr*, *Vangl2*, or *Vangl2*_Δ*NLS*_. Next, we generated mosaic organoids by combining *WT* primary basal cells with *WT* primary luminal cells (*WT/WT*) or *Vangl2 KD* primary luminal cells (*WT/KD*) that overexpress either control Scramble (*Scr* 1° LECs), *Vangl2* (*WT/KD + V2* 1° LECs), or *Vangl2*_Δ*NLS*_ (*WT/KD + V2*_Δ*NLS*_ 1° LECs), and grew them for 10 days in Matrigel (Figure 5A) [14]. We found that knocking down *Vangl2* in luminal cells resulted in larger organoids (Figure 5B,C). Although this increase in size was trending and not statistical, it was consistent with both our previously published results and our results in HC11 cells (Figure 4B,C) [9]. We also found that *Vangl2* reduced the size of *KD* organoids to that of *WT* organoids, whereas *Vangl2*_Δ*NLS*_ did not (Figure 5B,C). We harvested undifferentiated organoids to examine the expression of *Stat5a* and its target gene *Ccnd1*; we also harvested organoids at differentiation day 2 to examine the *Stat5a* target gene *Csn2*. In comparison to *WT*, we observed upregulation of *Stat5a* and *Csn2* with a trending increase in *Ccnd1* in *Vangl2 KD* luminal cells. We found the elevated expression of these genes in the *KD* was reduced by *Vangl2*, but not *Vangl2*_Δ*NLS*_ (Figure 5D,E), similar to our observations in HC11 acini (Figure 4B–F). Thus, in the absence of nuclear VANGL2, we observed upregulation of master lactogenic differentiation gene *Stat5a*, and its targets *Ccnd1* and *Csn2*, resulting in larger organoid growth and precocious differentiation. Taken together, our data present a role for nuclear VANGL2 in preventing lactogenic differentiation by binding to and repressing the *Stat5a* promoter.

## 4. Conclusions

Lactogenic differentiation generates mammary alveoli comprising luminal epithelial cells that express >1400 genes to generate the copious quantities of milk required to nourish offspring. About 30% of these genes are directly dependent on the STAT5A/B transcription factors that function downstream of progesterone and prolactin hormone signaling. The expression of many of these genes is driven by super-enhancers that rely on STAT5A/B to drive the huge increases (~1000 fold) in expression of milk genes such as *Wap* and *Csn2* [56,57]. Even though STAT5A/B are functionally redundant [45,58], STAT5A is more abundant in the mammary gland and has emerged as the master regulator of alveologenesis [40,45,55,56,59]. Even the transcription factor, ELF5, also considered a master regulator of mammary alveologenesis, depends on STAT5A to enhance its expression during pregnancy [46,47]. Indeed, the *Stat5a* locus contains a mammary specific autoregulatory enhancer to ensure the very high levels of STAT5A required for lactogenic differentiation [55]. Thus, there is an increasing understanding of the complex mechanisms required to ensure robust *Stat5a* expression during mammary alveologenesis [60], but less clear are the mechanisms that keep *Stat5a* expression in check to prevent precocious lactogenic differentiation. Several proteins that repress STAT5 transcriptional activity have been identified. These repressors either bind directly to STAT5A/B and interfere with DNA binding (e.g., SMRT or SHD1), or they interfere with STAT5A/B activation by preventing STAT5A/B phosphorylation (e.g., SOCS proteins) [61]. Here, we identify VANGL2 as a protein that represses *Stat5a* expression by binding directly to promoter 2 of *Stat5a*.

As a core PCP protein, VANGL2 is localized asymmetrically on the plasma membrane where it coordinates the orientation of cells within a tissue plane. Only more recently has VANGL2 been observed in membranous compartments within the cell, including the perinuclear space [4,5,6,7,8], and, here, we show VANGL2 in the nucleus. A previous study in zebrafish also identified the NLS motif in the C-terminal region of VANGL2 [49]. Using ectopic expression of a plasmid encoding only the VANGL2 C-terminal region, the authors found this portion localized to the nucleus, where it generated defects in convergent extension, a process driven by planar polarized cell intercalation, with higher efficiency than a control plasmid containing a mutated NLS. In contrast, our study of endogenous VANGL2 identifies localization of the full-length protein within the nucleus. Furthermore, we show that VANGL2 is recruited to numerous DNA motifs, including one in the *Stat5a* promoter, where our data support a role for it acting as a repressor that prevents *Stat5a* expression, thereby inhibiting lactogenic differentiation.

Prickle1 (Pk1) and Prickle2 (Pk2) are cytoplasmic PCP proteins that bind to and play key roles governing the asymmetric functions of VANGL proteins on the plasma membrane [62]. Interestingly, they have also been observed in the nucleus [63,64,65]. Pk1, also known as RE-1 silencing transcription factor (REST) interacting LIM domain protein (RILP), can be tethered to the membrane via post-translational farnesylation. This modification and phosphorylation on two sites by protein kinase A are required for Pk1/RILP nuclear localization, which, in turn, is necessary for the nuclear transport of the transcriptional repressor, REST [66,67]. A point mutation in Pk1/RILP that disrupts REST binding blocks the nuclear export of REST, generating a constitutively active repressor that silences a wide variety of genes, a loss that causes progressive myoclonus epilepsy syndrome [63]. It is unclear whether the nuclear functions of Pk1/RILP and Pk2 are related to their role in PCP. Similarly, we have not linked transcriptional repression mediated by VANGL2 to its role in regulating PCP, nor have we linked the nuclear functions of VANGL2 and Pk1/Pk2. Future studies will be required to further interrogate the relationship between PCP and the nuclear function of PCP proteins such as VANGL and Prickle.

Numerous full-length single transmembrane domain growth factor receptors and adhesion molecules have been identified in the nucleus [13]. Moreover, multi-transmembrane domain proteins, such as the tetraspanins (TSPANs) and seven transmembrane G-protein coupled receptors, have been located there as well [12,68]. Yet, the trafficking mechanisms governing the transport of transmembrane proteins into the nucleus are still incompletely understood. There are several models, including INTERNET (integral trafficking from the endoplasmic reticulum to the nuclear envelope transport) and NAE (nuclear envelope associated endosomes). The former relies on the continuity between endoplasmic reticulum membranes and the outer nuclear membrane to achieve nuclear incorporation, whereas the latter postulates a population of endosomes that is directed to and fuses with the outer nuclear membrane [69,70]. Once in the outer nuclear membrane, transmembrane proteins that contain an NLS in their cytosolic domain, such a VANGL2, can be bound by importins and transported through the nuclear pore into the inner nuclear membrane [71]. The final step, at least for single pass transmembrane receptors, is thought to be extraction from the membrane, followed by solubilization and stabilization, perhaps in conjunction with a chaperone protein, then release into the nucleoplasm. Other means to achieve solubilization are lipid modifications such as binding cholesterol as observed for TSPAN8 [68]. However, it is also possible that multi-transmembrane domain proteins remain within the inner nuclear membrane and function from this location.

Our studies have not addressed how VANGL2 is transported to the nucleus, but we find the entire protein, and not just a truncated C-terminal portion of it [49], in the nuclear soluble and insoluble fractions by biochemical fractionation of HC11 cells. Over differentiation, VANGL2 in the nuclear soluble fraction dramatically decreases while the nuclear insoluble fraction does not significantly decrease, suggesting that nuclear VANGL2 may be soluble in the nucleoplasm in addition to being embedded in the inner nuclear membrane. Furthermore, we find both N-terminal serine phosphorylated and non-phosphorylated forms of VANGL2 within the nuclear fraction. N-terminal serine phosphorylation of VANGL2 has been linked to its PCP role on the plasma membrane [33,35], whereas internalized VANGL2 in recycling endosomes and lysosomes does not appear to be phosphorylated [72]. Yet, phosphorylation regulates the nuclear trafficking of many transmembrane receptors including epidermal growth factor receptor (EGFR) and TSPAN8 [68,73], raising the possibility that some fraction of internalized VANGL2 is phosphorylated as a regulatory mechanism for nuclear translocation.

In conclusion, we have demonstrated the nuclear localization of VANGL2 is dependent on the differentiation state of the cells and found VANGL2 bound to DNA binding motifs in undifferentiated mammary HC11 cells, including one in the P2 promoter of the *Stat5a* gene. STAT5A is the master transcriptional regulator of lactogenic differentiation. In the absence of VANGL2, we observe upregulation of *Stat5a* and its downstream genes, as well as precocious differentiation of mammary epithelial cells. Taken together, our data demonstrate a novel function of VANGL2 as a transcriptional repressor regulating mammary lactogenic differentiation.

## Figures and Tables

**Figure 1 cells-13-00222-f001:**
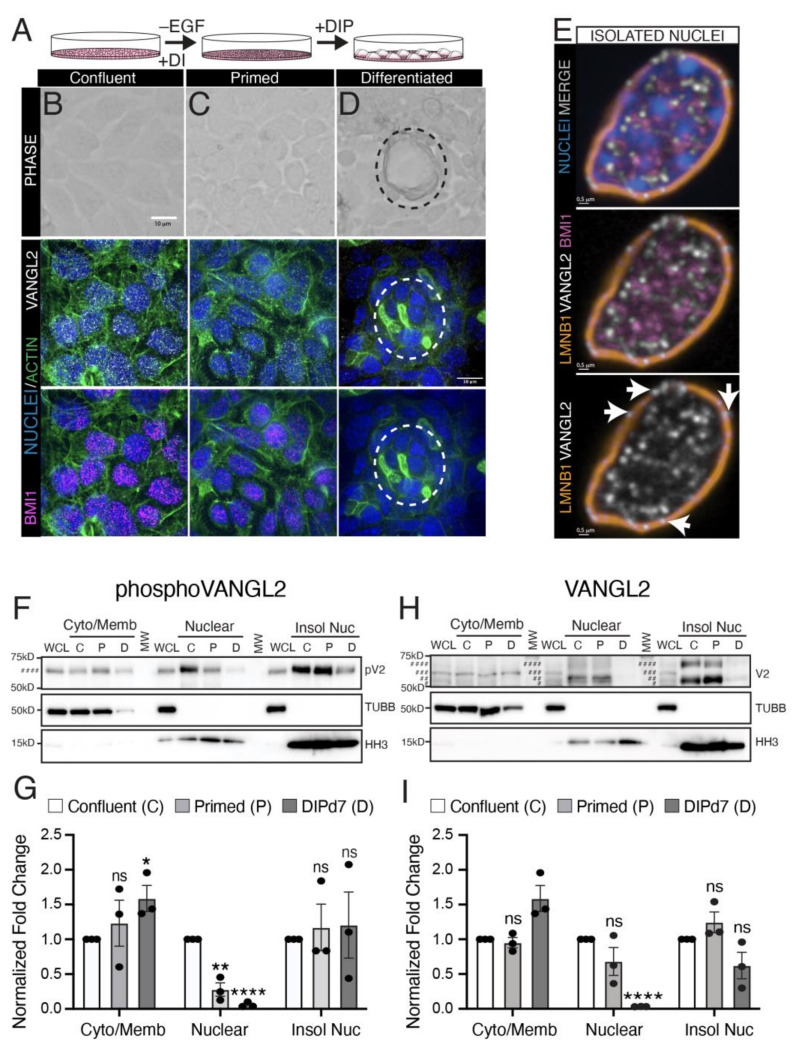
VANGL2 localizes to the nucleus in undifferentiated HC11 cells. (**A**) Cartoon illustration of the HC11 cell differentiation process. Cells were grown to confluence and cultured for 2 days, before priming for 18 h in the absence of epidermal growth factor (EGF) and presence of dexamethasone (**D**) and insulin (**I**). Then, the cultures were induced to differentiate by culturing in dexamethasone, insulin, and prolactin (DIP) for up to seven days. (**B**–**D**) Representative phase contrast (top panel) or immunofluorescence (bottom panels) micrographs of HC11 cells at different stages of differentiation: confluent (**B**), primed (**C**), and day 7 of differentiation (**D**). Immunofluorescence shows cells stained for VANGL2 (white), BMI1 (pink), and nuclei labeled with Hoechst (blue). Dashed circle indicates milk dome. Scale = 10 µm. (**E**) Representative immunofluorescent micrographs of nuclei isolated from confluent HC11 cells and stained for VANGL2 (white), BMI1 (pink), and Lamin B1 (LMNB1; orange). Nuclei labeled with Hoechst (blue). Arrows indicates VANGL2 puncta. Scale = 0.5 µm (**F**) Representative Western blot of phosphoVANGL2 (pV2) at different stages of differentiation: confluent (C), primed (P), and differentiated day 7 (D). HC11 whole cell lysates (WCL) were fractionated as follows: cytosolic/membrane (Cyto/Memb), soluble nuclear (Nuclear), and insoluble nuclear (Isol Nuc). Fractions were assessed using specific antibodies: ß-tubulin I (TUBB) for cytoplasmic/membrane fraction and histone H3 (HH3) for nuclear fraction. One form of VANGL2 was observed: hyperphosphorylated, ***. (**G**) Quantification of pVANGL2 relative to its loading controls of ß-tubulin I (TUBB) for cytoplasmic/membrane fraction and histone H3 (HH3) for nuclear fraction. PhosphoVANGL2 levels were normalized relative to confluent. (**H**) Representative Western blot of VANGL2 (V2) at different stages of differentiation: confluent (C), primed (P), and differentiated day 7 (D). HC11 whole cell lysates (WCL) were fractionated as described in (**F**). Four forms of VANGL2 were observed: unphosphorylated, #; hypophosphorylated, ##; phosphorylated, ###; and hyperphosphorylated, ####. (**I**) Quantification of VANGL2 relative to its loading controls as described in (**G**). VANGL2 levels were normalized relative to confluent. For (**G**,**I**), N = 3 biological replicates. Data are represented as mean ± S. E. M. *p*-values determined using unpaired *t*-test with Welch’s correction. *p* values: ns ≥ 0.05, * < 0.05, ** < 0.01, **** < 0.0001.

**Figure 2 cells-13-00222-f002:**
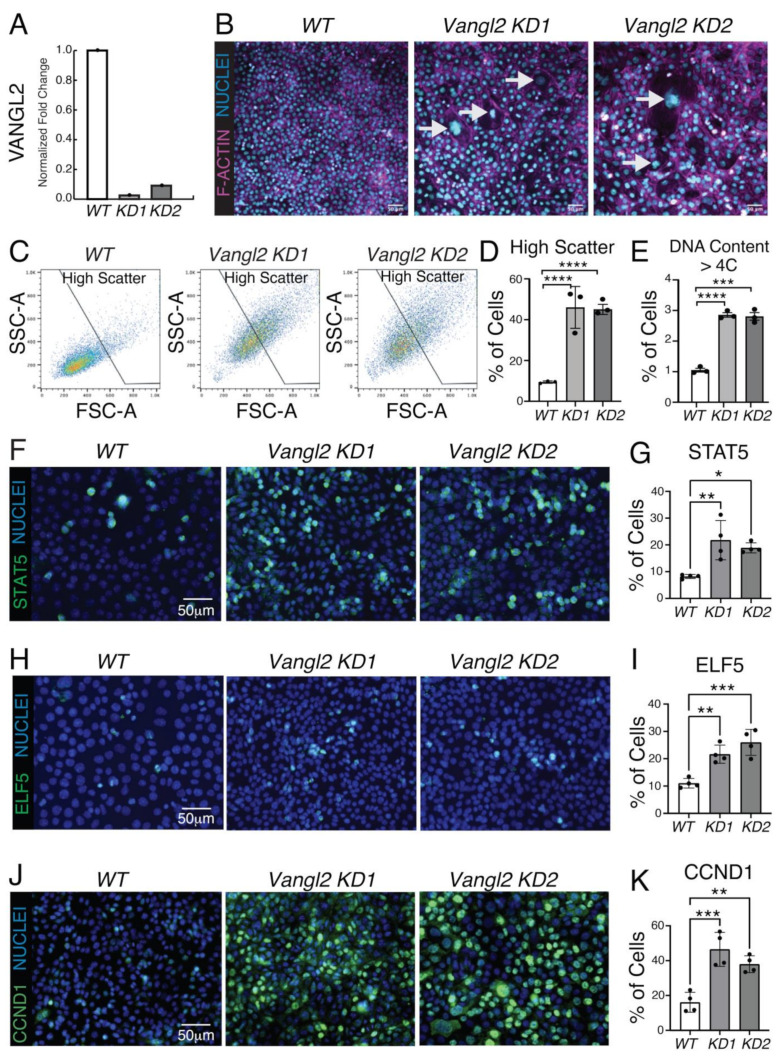
Undifferentiated *Vangl2* KO cells have increased polyploidy and express genes involved in lactogenic differentiation. (**A**) Western blot quantification of VANGL2 in *Vangl2 KD1* and *Vangl2 KD* 2 HC11 cells. (**B**) Representative immunofluorescent micrographs of *WT*, *Vangl2 KD1*, and *Vangl2 KD2* cells immunostained for F-actin (magenta). Nuclei labeled with Hoechst (blue). Arrows indicate large nuclei in the *KD* cells. Scale = 50 µm. (**C**) Representative FACS dot plots showing forward scatter (FSC-A) and side scatter (SSC-A) properties of *WT*, *Vangl2 KD1*, and *Vangl2 KD2* HC11 cells. (**D**) Percentage of *WT*, *Vangl2 KD1*, and *Vangl2 KD2* HC11 cells with high scatter properties. (**E**) Percentage of *WT*, *Vangl2 KD1*, and *Vangl2 KD2* HC11 cells with > 4C DNA content. (**F**) Representative immunofluorescence micrographs of *WT* cells, and *Vangl2 KD1* and *Vangl2 KD2* cells immunostained for STAT5 (green) with nuclei labeled with Hoechst (blue). Scale = 50 µm. (**G**) Quantification of the percentage of cells stained positive for STAT5. (**H**) Representative immunofluorescent micrographs of *WT* cells, and *Vangl2 KD1* and *Vangl2 KD2* cells immunostained for ELF5 (green) with nuclei stained by Hoechst (blue). Scale = 50 µm. (**I**) Quantification of the percentage of cells stained positive for ELF5. (**J**) Representative immunofluorescent micrographs of *WT*, *Vangl2 KD1*, and *Vangl2 KD2* cells immunostained for CCND1 (green) with nuclei labeled with Hoechst (blue). Scale = 50 µm. (**K**) Quantification of the percentage of cells stained positive for CCND1. For (**D**,**E**), N = 3 biological replicates and for (**G**,**I**,**K**), N = 4 biological replicates. Data are shown as mean ± SD. Two-tailed unpaired Student’s *t*-test. *p* values: *p* values: ns ≥ 0.05, * < 0.05, ** < 0.01, *** < 0.001, **** < 0.0001.

**Figure 3 cells-13-00222-f003:**
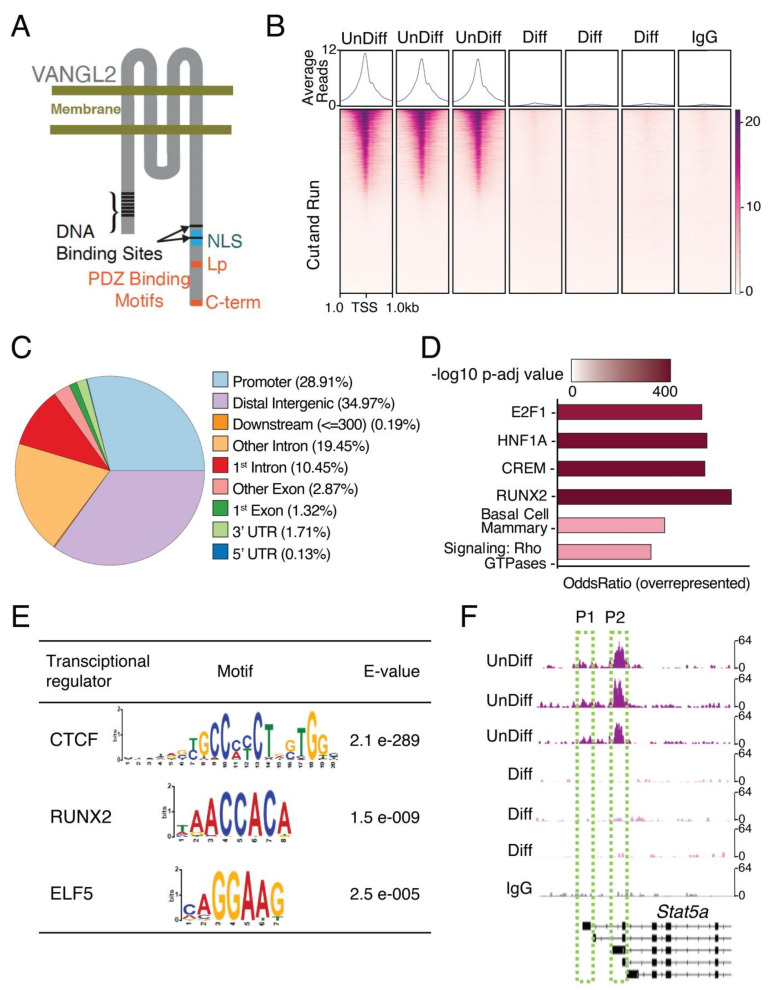
VANGL2 is recruited to distinct promoter sequences, including the *Stat5a* P2 promoter. (**A**) Cartoon illustration of VANGL2 showing DNA binding sites, nuclear localization signal (NLS), and two PDZ binding motifs, one containing the looptail (Lp) locus and another at the C-terminus (C-term). (**B**) Average plot (top) and heatmap (bottom) of VANGL2 CUT&RUN reads around the transcription start site (TSS ± 1 kb) from undifferentiated (unDiff) or differentiated (Diff) HC11 cells in triplicate, or of non-specific rabbit IgG CUT&RUN reads. The gradient purple color indicates high-to-low counts in the corresponding TSS regions. (**C**) Proportion of peaks called from VANGL2-enriched CUT&RUN reads corresponding to different genomic regions shown in color. (**D**) Enriched pathways of genes overrepresented in the undifferentiated HC11 cells (unDiff) as identified by VANGL2 CUT&RUN reads. The color gradient scale is logarithmic, with the darker colors representing higher *p*-values. E2F1, HNF1A, CREM, and RUNX2 were identified through ChIPSeq database, Basal Cell Mammary through Tabula Muris, and Signaling by Rho GTPases through REACTOME. (**E**) Motif analysis performed by MEME and STREME shows transcription factors motifs identified by VANGL2 CUT&RUN reads in undifferentiated HC11 cells (unDiff). The height of each letter is proportional to its frequency at that particular position. E-values estimate the expected number of motifs that would be found in a similarly sized set of random sequences. (**F**) Integrative Genomics Viewer (IGV) browser tracks showing VANGL2 binding in the promoter 1 (P1) and promoter 2 (P2) regions within the *Stat5a* gene locus in undifferentiated HC11 cells (unDiff), differentiated HC11 cells (Diff), or anti-rabbit IgG. All samples have the same scaling factor (0–64) for the *y*-axis.

**Figure 4 cells-13-00222-f004:**
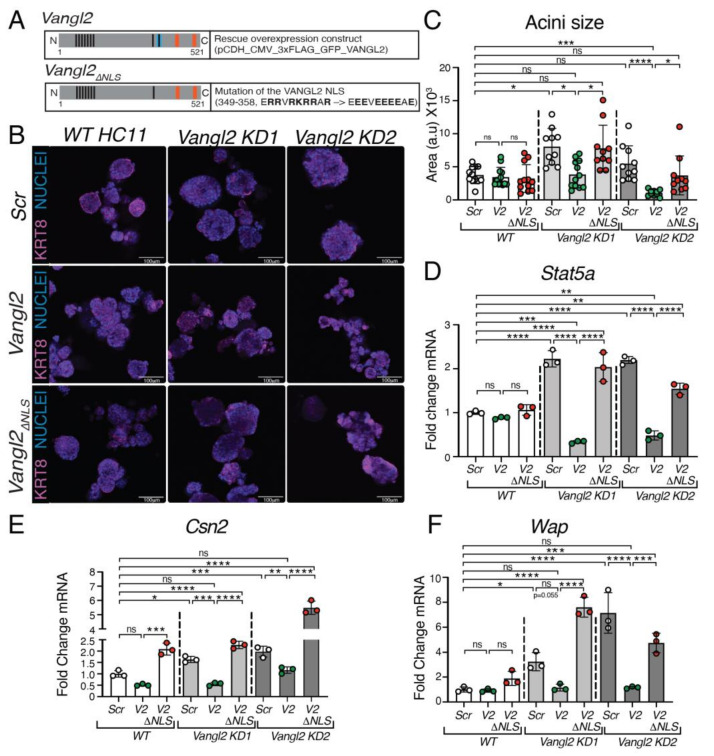
VANGL2 contains a nuclear localization signal that inhibits proliferation and acini formation. (**A**) Cartoon illustration showing *WT* and Δ*NLS Vangl2* lentiviral constructs. Black lines represent DNA binding sites, blue lines represent the nuclear localization signal (NLS) and orange lines represent the PDZ domains. (**B**) Representative immunofluorescence micrographs of acini grown in 3D Matrigel: *WT*, *Vangl2 KD1*, and *Vangl2 KD2* HC11 cells that overexpress either *WT Vangl2* or *Vangl2* Δ*NLS* lentiviral constructs. Acini were immunostained for KRT14 (pink) with nuclei labeled with Hoechst (blue). (**C**) Quantification of acini area after being grown for 5 days in 3D Matrigel: *WT*, *Vangl2 KD1*, and *Vang2 KD2* HC11 cells each overexpressing either *Scr* control, *WT Vangl2*, or Δ*NLS Vangl2* lentiviral constructs. (**D**) Quantification of the *Stat5a* expression in acini grown for 5 days in 3D Matrigel: *WT* HC11 cells, *Vangl2 KD1*, and *Vang2 KD2* cells each overexpressing either *Scr* control, *WT Vangl2*, or Δ*NLS Vangl2* lentiviral constructs. (**E**) Quantification of the *Csn2* expression in acini grown for 5 days and then differentiated for 2 days in 3D Matrigel: *WT*, *Vangl2 KD1*, and *Vang2 KD2* HC11 cells each overexpressing either *Scr* control, *WT Vangl2*, or Δ*NLS Vangl2* lentiviral constructs. (**F**) Quantification of the *Wap* expression in acini grown for 5 days and then differentiated for 2 days in 3D Matrigel: *WT*, *Vangl2 KD1*, and *Vang2 KD2* HC11 cells each overexpressing either *Scr* control, *WT Vangl2*, or Δ*NLS Vangl2* lentiviral constructs. For (**C**–**F**), N = 3 biological replicates. Data are shown as mean ± SD. For (**C**), data are from 10 organoids. Kruskal–Wallis test. For (**C**–**F**), one-way ANOVA Tukey’s test. *p* values: ns ≥ 0.05, * < 0.05, ** < 0.01, *** < 0.001, **** < 0.0001.

**Figure 5 cells-13-00222-f005:**
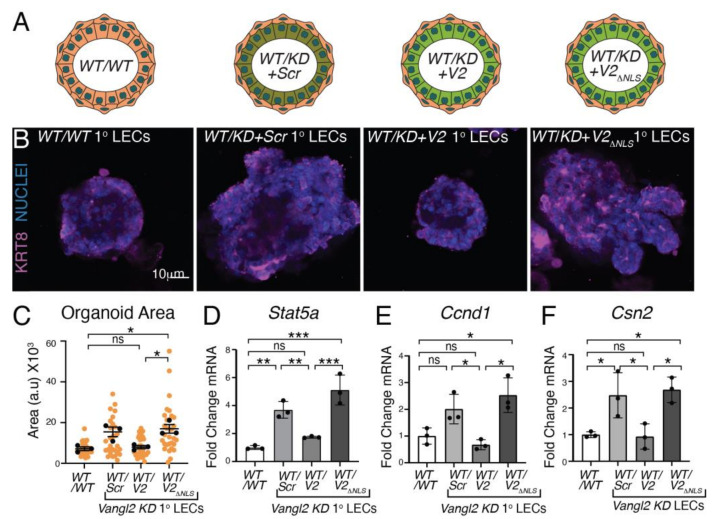
Nuclear function of VANGL2 is to repress genes that promote lactogenic differentiation. (**A**) Cartoon illustration showing mosaic organoids generated from primary murine mammary epithelial cells in which the basal compartment comprises *WT* primary mammary basal cells and the luminal compartment comprises either *WT* primary luminal cells (*WT*/*WT* 1° LECs) or *Vangl2KD* primary luminal cells overexpressing *Scr* control (*WT*/*KD* + *Scr* 1° LECs), *Vangl2KD* primary luminal cells overexpressing *Vangl2* (*WT*/*KD* + *V2* 1° LECs), or *Vangl2KD* primary luminal cells overexpressing *Vangl2*Δ*NLS* (*WT*/*KD* + *V2*_Δ*NLS*_) 1° LECs). (**B**) Representative immunofluorescence micrographs of mosaic organoids, as cartooned in (**A**), grown for 7 days and then differentiated for 2 days in 3D Matrigel. Organoids were immunostained for KRT8 (magenta), with nuclei labeled with Hoechst (blue). Scale = 10 µm. (**C**) Quantification of the area of mosaic organoids grown for 7 days and then differentiated for 2 days in 3D Matrigel. (**D**–**F**) Quantification of the *Stat5a* (**D**), *Ccnd1* (**E**), and *Csn2* (**F**) expression in organoids grown for 7 days and then differentiated for 2 days in 3D Matrigel. For (**C**), N = 3 biological replicates. Data are shown as mean of means ± SEM. Orange dots represent values from 30 organoids from N = 3 biological replicates (10 organoids/replicate). Black dots represent the mean area of each biological replicate. One-way ANOVA Tukey’s test. For (**D**–**F**), N = 3 biological replicates. Data are shown as mean ± SEM. One-way ANOVA Tukey’s test. *p* values: ns ≥ 0.05, * < 0.05, ** < 0.01, *** < 0.001.

## Data Availability

RNA-sequencing and CUT&RUN data generated in this study have been deposited in GEO: GSE253796.

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
