# Peer review of "Nuclear VANGL2 Inhibits Lactogenic Differentiation"

_cells, 2024, doi:10.3390/cells13030222_

Round 1

Reviewer 1 Report

Comments and Suggestions for Authors

This well-written manuscript explores the novel role of planar cell polarity proteins in lactogenic differentiation and tissue morphogenesis, restraining differentiation at the transcriptional level. It presents a number of important and novel findings that are of interest to the field of mammary gland biology and of interest to those in a wider field of epithelial biology. The observations that VANGL2 transcriptionally represses Stat5a, indicates that it is a major regulator of differentiation that had remained undiscovered and will open the field to new directions for both mammary gland biology but also other tissues where planar cell polarity is essential, which is virtually all epithelial organs.

The techniques are modern and highly appropriate for the experiments conducted. Both HC11, a cell line capable of differentiating in culture and primary mammary epithelial cells were used in the experiments, providing validity to the results.

Novel findings

VANGL2, a transmembrane protein, is transported to the nucleus and is dependent on the differentiation state of the cells

VANGL2 binds DNA in undifferentiated mammary epithelial cells, including the Stat5a gene.

Loss of Vangl2 upregulates Stat5a, Ccnd1 and Csn2,

Loss of Vangl2 resulted in larger acini and organoids,  and precocious diferentiation

Rescue experiments were very thorough, and included overexpression of Vangl2 and a nuclear localization signal mutant, which helped validate and also identify the molecular mechanism.

The variability between HC11 knockouts also reveals an important discovery for the field, highlighting the potential variability of the line, which is heavily relied upon.

Minor points

1.     Unless the journal rules are different, mouse genes should be italicized with only the first letter capitalized.

2.     The full name of BM1 should be presented.

3.     Asterisk are used to identify hyperphosphorylated VANGL2 on the WB but also statistical significance and should be different

4.     The mouse strain for the primary cells is not described or identified. Is it different than the HC11 origin?

This work provides interesting avenues moving forward, for example identifying what controls VANGL2 phosphorylation, and cross-talk with other pathways, which are out of scope of this manuscript.

Author Response

Response to Reviewers:
Reviewer 1:
This well-written manuscript explores the novel role of planar cell polarity proteins in lactogenic differentiation and tissue morphogenesis, restraining differentiation at the transcriptional level. It presents a number of important and novel findings that are of interest to the field of mammary gland biology and of interest to those in a wider field of epithelial biology. The observations that
VANGL2 transcriptionally represses Stat5a, indicates that it is a major regulator of differentiation that had remained undiscovered and will open the field to new directions for both mammary gland biology but also other tissues where planar cell polarity is essential, which is virtually all epithelial organs.

The techniques are modern and highly appropriate for the experiments conducted. Both HC11, a cell line capable of differentiating in culture and primary mammary epithelial cells were used in the experiments, providing validity to the results.
Novel findings:
• VANGL2, a transmembrane protein, is transported to the nucleus and is dependent on the differentiation state of the cells
• VANGL2 binds DNA in undifferentiated mammary epithelial cells, including the Stat5a gene.
• Loss of Vangl2 upregulates Stat5a, Ccnd1 and Csn2,
• Loss of Vangl2 resulted in larger acini and organoids, and precocious diferentiation
• Rescue experiments were very thorough, and included overexpression of Vangl2 and a nuclear localization signal mutant, which helped validate and also identify the molecular mechanism.
• The variability between HC11 knockouts also reveals an important discovery for the field, highlighting the potential variability of the line, which is heavily relied upon.
Thank you for this supportive review. We addressed the following points, which strengthened the manuscript.
Minor points:
1. Unless the journal rules are different, mouse genes should be italicized with only the first letter capitalized.
We have rechecked all the gene nomenclature. According to several journal and web sites (https://academic.oup.com/molehr/pages/Gene_And_Protein_Nomenclature;
https://www.biosciencewriters.com/Guidelines-for-Formatting-Gene-and-Protein-Names.aspx), mouse genes are italicized, with only the first letter in upper-case (e.g., Gfap). Proteins, however, are not italicized, and all letters are in upper-case (e.g., GFAP). We tried to adhere to these rules and would happy to change any instances where we failed.

2. The full name of BM1 should be presented.
We defined B cell-specific Moloney murine leukemia virus integration site 1 (BMI1) in the revised manuscript.

3. Asterisk are used to identify hyperphosphorylated VANGL2 on the WB but also statistical significance and should be different
We change the notation on the WB to the pound sign to indicate phosphorylation.
4. The mouse strain for the primary cells is not described or identified. Is it different than the HC11 origin?
HC11 cells were derived from the Comma-1D cell line 1, and the Comma-1D cell line was derived from the inbred BALB/c strain of mice during the middle of pregnancy

2. Our experiments shown in Figure 5 were done using the outbred CD1 strain of mice. We report this in our revised Methods.
This work provides interesting avenues moving forward, for example identifying what controls VANGL2 phosphorylation, and cross-talk with other pathways, which are out of scope of this manuscript.

Reviewer 2 Report

Comments and Suggestions for Authors

In the paper by Rubio et al., the authors convincingly demonstrate a role for the VANGL2 nuclear localization sequence in driving VANGL2 nuclear translocation and regulating lactogenic differentiation in HC11 cells and mammary gland explants. Overall, the authors convincingly demonstrate an impact of nVANGL2 on gene expression, lactogenic phenotype and PCP impact on gland organization and function. This will be an excellent contribution to the field. Recommended (but not required) before publication:

1.     It would be helpful to define the difference between confluence and priming when describing the results for figure 1.

2.     In figure 1, it is unclear the difference between F and H. it would be easier to understand if the pVANGL2 blot was simply shown above the total VANGL2 blot if this is the same experiment. There is no H described in the figure legend.

3.     It would be nice to add one piece of the supplementary data fig showing the relative knockdown efficiency of the 2 clones into figure 2 to easily compare to the phenotypic data.

Author Response

Reviewer 2:
Comments and Suggestions for Authors
In the paper by Rubio et al., the authors convincingly demonstrate a role for the VANGL2
nuclear localization sequence in driving VANGL2 nuclear translocation and regulating lactogenic
differentiation in HC11 cells and mammary gland explants. Overall, the authors convincingly
demonstrate an impact of nVANGL2 on gene expression, lactogenic phenotype and PCP
impact on gland organization and function. This will be an excellent contribution to the field.
Recommended (but not required) before publication:
1. It would be helpful to define the difference between confluence and priming when
describing the results for figure 1.
We provided a more detailed description both in the Results and the Methods for the growth and
differentiation of HC11 cells. Thank you for your suggestion.
Revised Results:
To explore a potential functional role for VANGL2 in the nucleus, we used HC11 cells because they offer
an in vitro model of mammary lactogenic differentiation. To facilitate differentiation, HC11 cells are
grown to confluence and maintained for two days before priming by withdrawing epidermal growth
factor. Differentiation is then induced by treating the cultures with prolactin, along with dexamethasone
and insulin, resulting in the formation of milk domes in five to seven days (Figure 1A-D) 2.
Revised Methods:
We purchased HC11 cells directly from ATCC(CRL-3062™), which were derived from mammary glands of
an inbred BALB/C strain mouse, just prior to initiating the studies. We expanded the line, froze numerous
vials and then thawed new, low passage cells to use for experiments and for generating the Vangl2 KD
lines. This cell line behaved as expected and differentiated over the published time course in response to
dexamethasone, insulin and prolactin. Undifferentiated HC11 cells were cultured in growing medium
(RPMI-1640; Thermo Fisher Scientific, 72400047), supplemented with 10% FBS, 5 μg/ml insulin (Millipore-
Sigma, I6634), 10 ng/ml epidermal growth factor (EGF; Peprotech, AF-100-15) and 1× AntiAnti (Thermo
Fisher Scientific, 15240112) at 37°C with 5% CO2. To differentiate cells, first cells were made competent by
growing them to confluency and then maintaining them in growing medium for 2 days. Next the cells
were primed by culturing them in priming medium [RPMI-1640 supplemented with 5% charcoalstripped-
FBS (Equitech Bio, SFBM31) 5 μg/ml insulin, 1 μM dexamethasone (Millipore-Sigma, D4902-1G)
and 1× Anti-Anti] for 18 h at 37°C with 5% CO2. To induce differentiation, primed HC11 cells were
cultured in DIP Medium [RPMI-1640, supplemented with 10% FBS, 1 μM dexamethasone, 5 μg/ml
insulin, 3 μg/ml Prolactin (NHPP, oPRL-21) and 1× anti-anti] at 37°C with 5% CO2.
2. In figure 1, it is unclear the difference between F and H. it would be easier to understand if
the pVANGL2 blot was simply shown above the total VANGL2 blot if this is the same
experiment. There is no H described in the figure legend.
These are different blots so we can’t put them together in one Figure. On the left (F), we show a
representative blot probed with anti-phospho-VANGL2 and, due to space constraints, we show
the quantification of n=3 blots below (G). On the right (H), we show a representative blot probed
with anti-VANGL2 and the quantification of n=3 blots below (I). In response to the Reviewer’s
comment, we made the figure less confusing by places labels (phospho-VANGL2 / VANGL2 at
the top of figures F and H. We also shortened the legend so the descriptions of (H) and (I) can
be read more easily. We submitted the full blots to the journal.
3. It would be nice to add one piece of the supplementary data fig showing the relative
knockdown efficiency of the 2 clones into figure 2 to easily compare to the phenotypic data.
Thank you for the suggestion. We added the requested panel to Figure 2.

Reviewer 3 Report

Comments and Suggestions for Authors

The manuscript by Rubio et al. describes work undertaken to look at the PCP protein Vangl2 in HC11 cells, which are a mouse mammary epithelial cell line. The authors combine manipulation of Vangl2 in vitro with RNA sequencing along with Cut and Run analysis of HC11 cells in 3 states of differentiation.

The manuscript focuses on a previously undiscovered localisation of Vangl2 to the nucleus. They identify a nuclear localisation in Vangl2 and investigate the consequences of Vangl2 knockdown and overexpression in HC11 cells and primary mammary organoids. 

The authors suggest that this work shows a novel function for Vangl2 as a transcriptional repressor.

Major comments

My major concern is almost all of the work presented in the manuscript is from looking at Vangl2 in the mouse mammary epithelial cell line, HC11. Although the abstract and other areas of the paper state that work has been done in HC11 cells and primary mammary organoids, the organoids are only partly comprised of primary cells.

The work shown in Fig4 is from acini comprised of HC11 cells. The work shown in Fig5, does show organoids but these are mosaic organoids comprised of primary basal cells with HC11 luminal cells. As far as I can tell from the methods and the legend for Fig. 5 all of the luminal cells are HC11 cells. If I have misread this and some are completely comprised of primary cells, then additional information should be provided to clarify this.

The potential identification of Vangl2 in the nucleus and its role inhibiting differentiation is important but it could just be an artifact of HC11 cells. It is critical for the authors to show Vangl2 nuclear organisation in at least one other mammary epithelial cell line and/or in primary mammary gland tissue. In both cells or mammary gland sections it should be shown that the nuclear localisation is present in undifferentiated but not differentiated cells (as shown for HC11 Cells).

The level of Vangl2 in HC11 cells and any other cells used in the study should be provided e.g. by qPCR. If the levels are low, then knockdown experiments are not meaningful.

Minor comments

Methods

-Please provide information about the HC11 cells

-add additional information about what the different mosaic organoids are comprised of, the current statement that organoids were prepared as previously described with references is not sufficient. 

- provide information about differentiation of HC11 cells- this is mentioned in section 3.1 of results but no information is provided in the methods.

-add information about the phospho-specific Vangl2 antibody in the methods.

-Fig.4 comes after Fig. 5 in the paper-these figures should be re-arranged.

-In Fig.5 C there are some clear outliers (in Scr and V2 deltaNLS groups) where the organoid area is way higher than the rest of the data points. Please assess whether these are true outliers and if so remove them from the data and re-calculate the organoid area and re-do the statistical analysis. 

-Images of KD2 combinations of acini should be shown in Fig.4B

Round 2

Reviewer 3 Report

Comments and Suggestions for Authors

Thank you to the authors for addressing my previous comments and adding some informative additional data.

The graph showing mRNA levels of Vangl2 in HC11 cells compared to embryonic stem cells is important to the work presented in the manuscript. This is currently just provided in the reviewers' response; please incorporate this data into the main manuscript and comment on whether/how this affects the overall interpretation of data in the discussion.

Author Response

Thank you for the suggestion. We have included the requested data in Supplementary Figure 1E.